# Therapeutic Potential of MRGPRX2 Inhibitors on Mast Cells

**DOI:** 10.3390/cells10112906

**Published:** 2021-10-27

**Authors:** Hiroyuki Ogasawara, Masato Noguchi

**Affiliations:** 1Pharmaceutical Frontier Research Laboratories, Central Pharmaceutical Research Institute, Japan Tobacco Inc., Yokohama 236-0004, Japan; masato.noguchi@adst.keio.ac.jp; 2Office of Research Development and Sponsored Projects, Shinanomachi Campus, Keio University, Tokyo 160-8582, Japan

**Keywords:** mast cell activation, MRGPRX2, MRGPRX2 inhibitor, neurogenic inflammation, type 2 inflammation, non-histaminergic itch, pseudoallergic reaction, host defense

## Abstract

Mast cells (MCs) act as primary effectors in inflammatory and allergic reactions by releasing intracellularly-stored inflammatory mediators in diseases. The two major pathways for MC activation are known to be immunoglobulin E (IgE)-dependent and -independent. Although IgE-dependent signaling is the main pathway to MC activation, IgE-independent pathways have also been found to serve pivotal roles in the pathophysiology of various inflammatory conditions. Recent studies have shown that human and mouse MCs express several regulatory receptors such as toll-like receptors (TLRs), CD48, C300a, and GPCRs, including mas-related GPCR-X2 (MRGPRX2). MRGPRX2 has been reported as a novel GPCR that is expressed in MCs activated by basic secretagogues, neurokinin peptides, host defense antimicrobial peptides, and small molecule compounds (e.g., neuromuscular blocking agents) and leads to MC degranulation and eicosanoids release under in vitro experimental condition. Functional analyses of MRGPRX2 and Mrgprb2 (mouse ortholog) indicate that MRGPRX2 is involved in MC hypersensitivity reactions causing neuroinflammation such as postoperative pain, type 2 inflammation, non-histaminergic itch, and drug-induced anaphylactic-like reactions. In this review, we discuss the roles in innate immunity through functional studies on MRGPRX2-mediated IgE-independent MC activation and also the therapeutic potential of MRGPRX2 inhibitors on allergic and inflammatory diseases.

## 1. Introduction

Mast cells (MCs) are at the forefront of exposure to environmental factors and are known to be important players and regulators of innate immune responses to pathogens [1,2]. MCs induce inflammation not only by direct exposure to pathogens, but also by sensing danger signals from surrounding tissues, including the epithelial cells, endothelial cells, and tissue-resident immune cells. [2,3]. MCs serve as sentinel innate immune cells, which store dense secretory granules containing mediators such as histamine, tumor necrosis factor α (TNF-α), serotonin, and also a wide range of MC-specific serine proteases bound to a proteoglycan core containing heparin and glycosaminoglycans, and are responsible for host resistance to bacteria, multicellular parasites, and xenobiotic toxins [4]. Upon MC activation, they rapidly release granules and produce de novo synthesized inflammatory mediators (cytokines and lipid mediators such as interleukin (IL)-4, -5, and -13), which are also involved in the continuation of allergic inflammation [5]. Thus, MCs have the ability to detect various danger signals transmitted by pathogens, tissues, and other immune cells and are responsible for modulating immune responses according to the characteristics of the stimuli received [1].

Two main pathways for MC activation are known: immunoglobulin E (IgE)-dependent and IgE-independent [5,6]. IgE-dependent MC activation is induced by the cross-linking of antigen with IgE bound to the high-affinity IgE receptor FcεRI [7]. Although IgE/FcεRI signaling is the main pathway for MC activation [8,9], the ability of mast cells to alter their responses to so many internal and external signals suggests that the various regulatory receptors such as toll-like receptors (TLRs), CD48, CD300a, lectin receptors, and GPCRs also act as regulators of these responses [10,11]. It is known that MCs are activated by basic secretagogues such as substance P (SP) and compound 48/80 (C48/80) [5,6], as well as through suppression of tumorigenicity-2 as an IL-1 family receptor for IL-33 and a receptor for thymic stromal lymphopoietin [12,13]. MC activation by basic secretagogues is mediated by the pertussis toxin (PTX)-sensitive G-proteins Gi2 and Gi3, which are known to be involved in phagocytosis. The MC activation by basic secretagogues has been reported to be mediated by Gi2 and Gi3, PTX-sensitive G-proteins, and then to activate phospholipase Cβ leading to exocytosis and mitogen-activated protein kinases (MAPKs), which induces synthesis and release of arachidonic acid metabolites [14]. It was also thought that basic secretagogues directly interact with G-proteins because of basic secretagogues directly activating G-proteins [15,16]; however, it was first reported that basic secretagogues activate connective tissue mast cells (CTMCs) via mas-related GPCR-X2 (MRGPRX2) [17]. MRGPRX2 was primarily presumed to be a GPCR expressed in the sensory neuron, but it highly expressed in human skin-derived and human cord blood-derived cultured MCs [17,18], and MRGPRX family members except MRGPRX2 were expressed in the neuron using transcriptome analysis [19]. Subsequently, cationic antimicrobial peptides and several US Food Drug Administration (FDA)-approved drugs involved in pseudoallergic reactions were also identified as MRGPRX2 agonists [20,21]. The identification of the MRGPRX2 mouse orthologue and its validation with knockout (KO) mice have significantly contributed to our understanding of the physiological roles of MRGPRX2 in vivo.

In this review, we will firstly outline the characteristics and downstream signaling of MRGPRX2, a very unique GPCR without parallel, and secondly discuss how MRGPRX2 activation may contribute to the etiology of neurogenic inflammation and pain, type 2 inflammatory reactions such as atopic dermatitis (AD), chronic urticaria (CU), allergic contact dermatitis (ACD), non-histaminergic itch, and small molecule compound-induced pseudoallergy. Finally, we propose that inhibiting the inappropriate activation of MRGPRX2 in MCs may lead to new therapeutic approaches for many allergic and inflammatory diseases.

## 2. MRGPRX2

*MRGPRs* were cloned as a GPCR family with 35% homology to the proto-oncogene *MAS1* gene and were reported to be sensory neuron-specific GPCR [22]. In humans, there are four MRGPRX families (*MRGPRX1-X4*) and *MrgprD-H* [23,24], but there are 22 *MrgprA* genes, 11 *MrgprB* genes, and 13 *MrgprC* genes in mice. Within the MRGPR family, *proenkephalin A* gene products such as bovine adrenal medulla (8–22) peptide were first identified as ligands for MRGPRX1 [22], followed by the identification of corstatin-14 (CST-14) as a ligand for MRGPRX2 [25]. In 2005, it was reported that PAMP-12, somatostatin, neuropeptide FF, oxytocin and SP are also ligands for MRGPRX2 [26]. The physiological function of MRGPRX2 has initially been proposed to be the involvement of nociception and catecholamine secretion from the adrenal gland, in dorsal root ganglia and adrenal chromophilic cells, based on the tissue distribution of *MRGPRX2* gene expression [26]. However, the canonical receptors for the peptides identified as ligands for MRGPRX2 are expressed in the neurons. The affinities of CST-14 and SP for canonical receptors (somatostatin receptor and neurokinin-1 receptor (NK-1)) are on the order of nM, whereas MRGPRX2 reacts with ligands on the order of μM [26,27,28]. Because of the large discrepancy in the ligand affinity, it was difficult to compare the effects of MRGPRX2 with those of other canonical receptors. Subsequently, it has been reported that MRGPRX2 is highly expressed in CTMCs and induces degranulation of human CTMCs by recognizing basic peptides such as mastoparan, somatostatin, SP, C48/80, VIP, and pituitary adenylate cyclase-activating polypeptide (PACAP) (6–27), which are known as basic secretagogues [17,29]. Furthermore, human β-defensin (hBD)2 and hBD3, as well as cathelicidin LL-37, were reported to induce MC degranulation via MRGPRX2 [21,30]. In addition, it was successively reported that the FDA-approved peptidermic drugs and small-molecule drugs (e.g., morphine and neuromuscular blocking agents) induced systemic pseudoallergic and anaphylactic reactions via MRGPRX2 [20]. The enzymatic degradation products of intracellular proteins released after cytotoxicity or cell death, such as chaperonin-10 (1–20), also activate MCs as MRGPRX2 ligands, and thus, MRGPRX2 could play a role in detecting danger signals after cytotoxicity [31]. The peculiar feature of MRGPRX2 is the recognition of a wide range of basic amino acids and basic low-molecular-weight compounds without amino acid sequence motifs, but MRGPRX2 is not a GPCR that indiscriminately recognizes peptides with a basic charge [32]. In addition, many small molecule agonists of MRGPRX2 are highly negatively charged compounds, but fluoroquinolones (pH 7.4), bipolar compounds containing both negative and positive charges, also activate MRGPRX2 [33]. This implies that a simple basicity alone cannot explain the atypical characteristics of MRGPRX2 ligands. Fujisawa et al. reported that full-length major basic protein (MBP) also employs MC activation via MRGPRX2 [18], while tryptase fragment peptides of MBP also activate MCs via MRGPRX2 [32]. Based on the crystal structure of full-length MBP, the MRGPRX2 activating regions corresponding to the MBP fragments are not exposed on the surface [34,35], suggesting that MRGPRX2 recognizes full-length MBP in a different mode from the MBP fragment. Since MRGPRX2 ligands have little common sequence regularity as peptide ligands and emerge as a wide range of molecules from small molecules to full-length proteins, MRGPRX2 could be considered a very inimitable GPCR with a complex ligand recognition system.

Mrgpra1 and Mrgprb2 have been reported as mouse orthologs that recognize the MRGPRX2 ligand SP [20,36], with *Mrgpra1* being expressed in the neurons and *Mrgprb2* in MCs [36]. Neurons have an NK-1 receptor of canonical SP receptors and Mrgpra1 also responds to SP [34], so it can be assumed that Mrgpra1 covers the role of MRGPRX2 in the neurons. The mouse and rat orthologues, *Mrgprb2* and *Mrgprb3*, which are expressed in MCs as observed in human MRGPRX2, recognize SP, hBDs, and pseudoallergic inducers [17,20,37]. Therefore, Mrgprb2 and Mrgprb3 have contributed greatly to the progress of research on the biological functions of MRGPRX2 expressed in MCs. Although Mrgprb2 covers most MRGPRX2 ligands to a large extent among MRGPRX2 orthologs in rodents, there are ligands with large differences in their action concentrations when comparing ligand action concentrations between Mrgprb2 and MRGPRX2 [20]. This can be attributed to the low amino acid sequence identity (45–65%) between the rodent MRGPR family and human MRGPRX2. Since the number of rodent Mrgpr family members is higher than that of human MRGPRX family members, it remains possible that Mrgpr family members other than Mrgprb2 are partially responsible for MRGPRX2 action in MCs. The canine GPCR gene cluster is more similar to humans than to rodents [38]. The canine MRGPRX2 orthologue has 62% sequence homology to human MRGPRX2, but the canine MRGPRX2 is more responsive to C48/80, SP, and fluoroquinolones than human MRGPRX2 [39]. Therefore, to understand MRGPRX2 function in vivo using experimental animal models requires consideration of the species differences.

It is generally known that GPCR ligands act on the hydrophobic pocket formed by the extracellular loops and the adjacent transmembrane regions of the GPCRs, converting them into cellular responses via G-proteins, β-arrestins, and other downstream effectors [40]. Furthermore, it has been shown that naturally occurring missense MRGPRX2 variants (G165E, D184H, W243R, and H259Y) affecting mast cell degranulation are located in amino acids located at the transmembrane helixes of the receptor [40]. MRGPRX2 is a PTX-sensitive Gi-coupled GPCR that rapidly induces intracellular Ca^2+^ mobilization and suppresses forskolin-induced intracellular cAMP elevation after MRGPRX2 ligands stimulation (Figure 1) [17,41,42]. This is consistent with the fact that basic secretagogues, ligands for MRGPRX2, have been reported to activate MCs via Gi2 and Gi3 [14]. Ligand stimulation of MRGPRX2 induces exocytosis via the phospholipase Cβ/calcineurin pathway after intracellular Ca^2+^ mobilization in MCs [14,42]. Store-operated Ca^2+^ entry via stromal interaction molecule 1 could promote MRGPRX2-induced human MC response and Mrgprb2-dependent inflammation in a mouse model [43], supporting that the Ca^2+^ mobilizing mechanism is an important signal for the physiological function of MRGPRX2. It is also known that phosphatidylinositol 3-kinase (PI3K)/AKT, MAPKs such as c-Jun N-terminal kinase (JNK), extracellular signal-regulated kinase 1/2 (Erk1/2), p38), and nuclear factor-κB (NF-κB) are involved in cytokine production from MCs during allergy [44,45,46,47]. MRGPRX2 ligands SP and LL-37 activate downstream signals of MRGPRX2, such as Erk1/2, JNK, p38, and PI3K/AKT, to produce TNF-α, IL-6, IL-8, and IL-13 [48,49,50], and LL-37 is involved in cytokine production by activating the NF-κB pathway through MRGPRX2 [39,51]. IL-33, a member of the IL-1 family, induces cytokine production in MCs through its downstream signals NF-κB and p38/JNK MAPK, and efficiently co-operates with MRGPRX2-mediated MC activation in the production of IL-8, TNF-α, chemokine ligand (CCL) 1, and CCL2 [52]. These studies indicate that NF-κB and p38/JNK signaling plays an important role in MRGPRX2-mediated MC cytokine production. Activation of MCs results in the release of stored inflammation-related mediators such as histamine and de novo synthesis of lipid-derived substances [5]. MRGPRX2 ligands SP, MBP protein, MBP and ECP enzyme fragments, and angiogenic host defense peptide AG-30/5C, induce de novo synthesis of prostaglandins in MCs [18,42,51,53]. Downstream signaling of MRGPRX2, Erk1/2, is involved in the induction of PGD2 de novo synthesis in MCs by SP [42]. In addition to G-protein signaling, C48/80 or SP recruits β-arrestin 1 via MRGPRX2 to internalize and desensitize MRGPRX2 at the plasma membrane [54,55] and to activate the ERK signaling pathway [40]. On the other hand, another MRGPRX2 ligand, AG-30/5C or icatibant, does not internalize MRGPRX2 [56]. Thus, MRGPRX2 ligands can be divided into the balanced ligands inducing both G-protein and β-arrestin signals and the biased ligands not inducing β-arrestin signals (Figure 1). At sites of microbial infection, mediators released from MC promote migration of dendritic cells, which eventually migrate into draining lymph nodes [57]. The lack of functional desensitization and internalization of MRGPRX2 caused by the biased ligands may enhance immune modulation. It has been reported that the phosphorylation of Tyr279 at the cytoplasmic end of the TM7 domain is important for the induction of MRGPRX2 internalization by the balanced ligand, SP [55]. In the future, the full picture of the biological effects of the different mechanisms of MRGPRX2 activation by the balanced and biased ligands will be clarified.

## 3. Neurogenic Inflammation in Postoperative Pain and Migraine

MCs are predominantly located in close proximity to peripheral nerve endings compared to other innate immune cells, making them the first cells to respond to sensory nerve activation. MCs are also involved in the mobilization of a variety of innate immune cells, furthering the inflammatory cascade and sensitization of peripheral afferents [58]. The crosstalk between the neurons and MCs has been implicated in the pathogenesis of postoperative pain and migraine [59,60,61].

Postoperative pain has both inflammatory and neuropathic components. Effective postoperative analgesia and sedation are needed to increase patient satisfaction and decrease postoperative morbidity and mortality. [62]. In peripheral tissues after surgical incision, the inflammatory response is maintained by inflammatory mediators from resident immune cells, including MCs [63]. Among the inflammatory mediators released from MCs, histamine and serotonin have been shown to cause postoperative nociception [60]. Inhibition of MC degranulation suppressed mechanical allodynia and spontaneous pain in a mouse model of postoperative pain, and stabilization of MCs has been reported to suppress nociceptive responses [60,64]. Postoperative blood SP is elevated [65]. In these circumstances, Green et al. reported that Mrgprb2-KO mice reduced postoperative pain and thermal hyperalgesia. Comparing thermal and mechanical hypersensitivity after incisional surgery between Mrgprb2-KO mice and wild-type controls, the expression of activating transcription factor 3, an injury indicator in dorsal root ganglion (DRG) cells, was reduced in Mrgprb2 KO mice [66]. These findings indicate that Mrgprb2 receptors on MCs are associated with pain and neuronal activation. The *Mrgprb2* receptor is expressed on MCs, but not expressed on DRG neurons. Mrgprb2-KO mice implied that SP-induced infiltration of other immune cells (leukocytes, neutrophils, and monocytes) was suppressed in Mrgprb2-KO mice and also the effect was NK-1-independent [66]. Therefore, these findings suggest that Mrgprb2 in MCs plays an important role in connecting the innate immune response to the nervous system. Although all factors mediating postoperative pain have not been identified, MRGPRX2, which can recognize multiple neurogenic peptides, appears to be deeply involved in neuron-MC crosstalk in postoperative pain. Despite significant progress in the study of the neurobiology of pain and the previous recognition of the undesirable effects of opioids, there has been no progress in the development of a new class of pain medications to replace opioids [67]. The fact that NK-1 antagonists were effective in preclinical studies but failed to show efficacy in multiple pain conditions [68] has led to skepticism about their preclinical results on nociceptive mechanisms and their efficacy in human pain. Pain mechanisms based on MRGPRX2-mediated neurogenic inflammation induction may suggest a potential pathway to new therapeutic agents.

Migraine, a transient headache, is thought to be caused by activation of the trigeminal nerves, consisting of the neurons innervating the cerebral vessels whose cell bodies are located in the trigeminal ganglion. Trigeminal afferents lead to the release of vasoactive peptides, causing neurogenic inflammation [69]. Chronic stress, dietary habits, hormonal fluctuations, and cortical spreading depression cause neurogenic inflammatory conditions in the intracranial meninges, resulting in the release of neurotransmitter and vasoactive neuropeptides such as SP and PACAP from nerve endings. This results in degranulation of MCs around the dura mater, increased vascular permeability, plasma protein efflux, and vasodilation. These reactions cause a spreading inflammatory response in adjacent tissues and induces orthostatic conduction along the trigeminal nerve fibers to register pain [17,61,70,71]. MCs indigenous to the meninges are thought to play an important role in triggering neurogenic inflammation of the meninges by producing cytokines and prostaglandins through SP and PACAP [61,72]. PAC1, the canonical receptor for PACAP, is not expressed on MCs, and antagonists of NK-1 did not suppress migraine [61]. SP is elevated during migraine headache [73]. In addition, the recent development of several innovative experimental migraine models has provided suggestive evidence of the involvement of SP in migraine headache [74]. This circumstantial evidence enables us to imagine that MRGPRX2 in MC may be deeply involved in the induction of neurogenic inflammation in migraine. It should be clarified using animal models such as Mrgprb2-KO mice. Clarifying the pain mechanism based on MRGPRX2-mediated neurogenic inflammatory induction could be a pathway to create new preventive therapeutics.

## 4. Type 2 Inflammation

### 4.1. Atopic Dermatitis (AD)

AD or eczema is a chronic, recurrent, itchy, inflammatory skin disease. The incidence of this disease has increased over the past 30 years, with 10–20% of children affected [75]. Although the etiology of this disease is not fully understood, it is multifactorial and is presumably caused by a complex interaction of genetic and environmental factors. Environmental allergens result in the abnormal activation of type 2 immunity and impair the barrier function of the dry, itchy skin, which is a typical lesion of AD. In addition, mechanical damage from scratching aggravates the lesion site [76]. Thus, the immune dysregulation due to Th2-dominant inflammation and MC sensitization, or a defect in skin barrier function, is believed to be the primary cause of this disease [77,78]. MCs play an important role in the pathogenesis of AD, an inflammatory disease accompanied by pruritus, because increased numbers of activated MCs and eosinophils known as neurogenic inflammation inducers have been reported in the skin of AD lesions [79]. Additionally, clinical studies have shown that large amounts of the cationic neuropeptide SP are detected in the serum of AD patients and that the amount of these neuropeptides correlates with the severity of the disease [80,81,82]. The neuropeptides released from the sensory neuron network (nociceptors) in the skin would be involved in the worsening of AD symptoms by promoting itching and scratching behavior of the lesion [83]. The SP precursor gene *TAC1* and the temperature-sensitive ion channel *TRPV1* are expressed in the neuropeptide-producing nociceptors [84]. Serhan et al. reported that the sensitizing antigen directly activated the neuropeptide-producing nociceptors to release SP by using an AD mouse model sensitized with extracts of house dust mites and Staphylococcus aureus (extracts of the strain Dermatophagoides farinae and staphylococcal enterotoxins B (SEB)) [84]. When the same AD model was evaluated using Mrgprb2-inactivated mutant mice, dermatitis induction (histological abnormalities of the skin, infiltration with eosinophils or neutrophils, specific IgE production against sensitizing antigens, and disruption of the barrier structure of the skin) was not observed [84]. Furthermore, the skin tissue damage induced after sensory nerve activation via TRPV1 stimulation was suppressed in Mrgprb2 mutant mice [83]. This model recapitulates a moderate to severe AD-like disease, with pathological features both of an exacerbated type 2 immune response and a global gene expression pattern that is statistically similar to human AD [84]. SEB used as one of the antigens is a superantigen that activates Th2 cells, and co-sensitization with Dermatophagoides farinae extract worsened the dermatitis and lesions more than SEB alone [84]. This result strongly suggests that MRGPRX2 in MCs may play a pivotal role in the progression to allergic skin diseases associated with type 2 immunity. Infiltration of eosinophils in AD lesions has been observed, suggesting that eosinophils are involved in AD progression as well as MCs [79]. MBP, an eosinophil granule from eosinophils that activates MRGPRX2, has been detected in AD lesions [85]. MRGPRX2 has been reported to be activated by MBP and the tryptase fragment of MBP [18,32]. These results suggest that MRGPRX2-mediated MC activation by the activated eosinophils infiltrating into the skin lesions may be involved in further exacerbation of allergic skin diseases. When IL-4 weakens MRGPRX2 signaling in skin MCs, MRGPRX2 could act as a bystander as well as FcεRI in Th2-dependent diseases [86].

### 4.2. Chronic Urticaria (CU)

In CU, in which wheals occur daily over the entire body for at least 6 weeks, the release of various vasoactive autacoid mediators, proteases, chemokines, and cytokines from MCs plays an important role in inducing the wheal response [87]. Approximately 40% of patients with chronic spontaneous urticaria have circulating IgE or FcεR1-recognizing IgG antibodies [88], and 30–50% of patients with CU express high levels of FcεRI [18]. For around half the patients with CU the condition has been implicated in IgE and autoimmune mechanisms, and the remainder are considered to have idiopathic CU [89]. On the other hand, it has been reported that there is no change in the number of mast cells in the lesions, but there is a higher percentage of MRGPRX2-positive mast cells in CU patients compared to healthy subjects [18]. In addition, intradermal injection of neuropeptides such as MRGPRX2 ligand SP and vasoactive intestinal peptide (VIP) enhances the wheal response in CU patients in comparison with healthy subjects [90,91]. Compared to healthy individuals, MRGPRX2 expressing mast cells and MRGPRX2 ligands such as SP in serum derived from CU patients are upregulated [90]. Similar to AD lesions, infiltration with eosinophils [92] and prominent deposition of MBP and eosinophil cationic protein (ECP) have been reported in CU lesions [93,94]. MBP and tryptase fragments of MBP and ECP activate MRGPRX2 to activate mast cells [18,32]. These reports suggest the involvement of MRGPRX2 in mast cells in CU lesions.

### 4.3. Allergic Contact Dermatitis (ACD)

ACD is considered to be a skin inflammation caused by repeated skin exposure to contact allergens or non-protein chemicals called haptens [95,96]. ACD causes skin inflammation when CD8^+^ effector T cells, which are initiated in the lymphoid organs during the sensitization phase, are recruited to the skin after re-exposure to the haptens. Since CD4^+^ regulatory T cells are involved in the suppression of CD8^+^ effector T cells, ACD is considered to be a disruption of cutaneous immune tolerance to haptens [95,96]. On the other hand, recent studies on the pathophysiology of ACD have also reported that the production of cytokines and chemokines by MC activation plays a major role in the magnitude of the inflammatory response mediated by CD8^+^ T cells [95]. Mrgprb2-KO mice have been reported to have significantly reduced inflammation, including itching in various models of allergic contact dermatitis [97,98,99], suggesting that MRGPRX2 is involved in the development of ACD as well as atopic dermatitis.

### 4.4. Rheumatoid Arthritis (RA)

In patients with RA, an increased number of activated MCs is observed in synovial tissue and fluid, and MC activation has been linked to the potentiation of inflammation in the pathogenesis of RA [100,101,102]. Synovial tissue in RA contains stem cell factor (SCF), which is important for MC survival, as well as cytokines such as IL-3 and IL-4. These mediators have been shown to induce MC proliferation, and SCF and TGFβ have been shown to induce MC recruitment [103]. The synovial MCs are activated not only by the activation of FcεRI, FcγRI, and FcγRII, but also by the neuropeptide SP [104]. The levels of neuropeptides such as SP and somatostatin are increased in synovial fluid derived from RA patients [105], and SP positive nerve fibers are also found in the synovium [106]. SP is localized around the cell membrane on MCs in synovium from RA patients. Cultured synovium-derived MC activation of releasing histamine and tryptase via SP stimulation is diminished by silencing the expression level of MRGPRX2 mRNA [104,107]. Histamine and tryptase released from MCs are known to increase vascular permeability and induce proliferation of synovial fibroblasts [108]. These findings suggest that the neurogenic inflammation via MRGPRX2-mediated MCs activation is involved in the pathogenesis of RA.

### 4.5. Ulcerative Colitis (UC)

It is widely accepted that environmental and host factors are intricately intertwined in the pathogenesis of UC, and events such as disruption of the mucosal barrier, changes in the healthy balance of the intestinal microbiota, and the abnormal stimulation of the intestinal immune response trigger the onset of the disease [109]. Substantial numbers of MCs have been noted in patients who have been diagnosed with UC [110]. UC is also thought to be associated with MC hyperplasia and humoral activity [111]. In a recent report, the expression of adrenomedullin, another MRGPRX2 ligand, is up-regulated in inflamed UC containing activated fibroblasts and epithelial cells. MC activation by adrenomedullin is more frequently detected in the gastrointestinal mucosal lamina propria derived from inflamed UC [112]. It is known that nerve fibers containing SP become thicker in the colonic mucosa of UC patients [113] and that the amount of SP in the rectum of UC patients is increased [114]. These findings suggest that the MRGPRX2 ligands are released from stromal cells and nerve cells via stimulation of the gastrointestinal mucosal layer and that MRGPRX2-mediated MC activation is involved in the mechanism of UC pathogenesis.

## 5. Non-Histaminergic Itch

Histamine is considered the classic itch substance, but most types of chronic itch are not ameliorated by antihistamines, so that non-histaminergic and neural mechanisms in chronic itch conditions have been studied [115,116]. As a non-histaminergic itch inducer, tryptase released from MCs is thought to be a major non-histaminergic itch mediator in diseases such as atopic dermatitis and ACD, activating PAR2 in afferent neurons to transmit itch [98,117]. It has been reported that the MC activation mechanisms of IgE and SP stimulation show different granule secretion patterns in space and time. MC stimulation with MRGPRX2 agonists rapidly activates them to secrete small, relatively spherical granule structures, whereas MC stimulation with anti-IgE increases the temporal gap between signaling and secretion, resulting in the secretion of slower, larger, and more heterogeneously shaped granule structures [118]. The ratio of contents in the secretory granules also differs between IgE and MRGPRX2 stimulations: more histamine and serotonin with IgE stimulation, but more tryptase with MRGPRX2 stimulation [97]. It has been reported that the different patterns between IgE and SP stimulation observed on MC degranulation are related to the signaling mechanism and that IKK-β inhibition during IgE-dependent stimulation changes to the similar degranulation pattern induced by SP [118]. IgE-dependent and SP-dependent activation also showed different patterns in degranulation of mouse MCs, respectively. Mrgprb2-KO mice showed attenuated itch in the ACD model [97]. These results suggest that MRGPRX2 contributes to the non-histaminergic itch. However, itching was not completely suppressed in this model [97], and it is possible that mechanisms other than MRGPRX2, such as IL-31, endothelin-1, and voltage-gated sodium channel (NaV), may be involved in non-histaminergic itching [119,120,121].

## 6. Drug-Induced Pseudoallergic Reaction

Non-allergic drug hypersensitivity, one of the specific side effects, is also called pseudoallergic reaction or anaphylactoid reaction [122]. Pseudoallergic reactions occur as a result of MC activation by drugs such as peptidic drugs, neuromuscular blocking agents, fluoroquinolones, vancomycin, and radiographic contrast agents [123,124]. These drugs have been reported to induce more serious consequences such as hypotension and shock-like syndromes, which are adverse events that lead to serious and potentially life-threatening treatment outcomes [125]. Pseudoallergic reactions to these drugs are thought to be IgE-dependent or IgE-independent responses [126]. Since 85% of the patients who had pseudoallergic reactions to neuromuscular blocking agents were not previously treated with neuromuscular blocking agents [127], IgE-independent MC activation by neuromuscular blocking agents may be a factor causing pseudoallergic reactions. It has been reported that the pseudo-allergic reactions to several FDA-approved drugs were from MRGPRX2-mediated MC activation [20,128,129]. Subsequently, clozapine, an antipsychotic drug, and codeine have been reported to cause pseudoallergic reactions via MRGPRX2 [54,130]. Usually, nonclinical toxicity studies for drug candidates are conducted using two animal species, rodents (rats or mice) and non-rodents (monkeys, dogs, or pigs). It is sometimes difficult to detect drug-induced cutaneous reactions suggestive of drug hypersensitivity in rats or mice. This is because rats or mice are tolerant to immune-mediated drug reactions due to their immune tolerance or high detoxification capacity of reactive metabolites [131]. In veterinary medicine, drug-induced cutaneous reactions also occur fairly commonly in dogs and some of these reactions are similar to those observed in humans [132], whereas there are some dog-specific cutaneous reactions, and their clinical relevance is considered to be low [133]. Morphine, fluoroquinolone, and C48/80, all of which also act as MRGPRX2 ligands, cause a rapid increase in histamine levels in dogs [133,134,135]. Many molecules involved in danger signals were also found to bind to toll-like receptors, which act as an alternative to pathogen-associated molecular patterns for activating antigen presenting cells and co-stimulating T cells to initiate an immune response [136]. T-cell checkpoint factors have also been suggested to be involved in drug-induced pseudoallergic reactions [137]. These factors do not fully explain the species differences and the above responses. Since it is difficult to identify all drugs that may cause the drug-induced pseudoallergic reactions in preclinical studies, symptomatic agents aiming to eliminate sudden drug-induced pseudoallergic reactions would be required in order to obtain the maximum therapeutic effect of the main action. From the viewpoint of modifying side-effects, it is necessary to develop MRGPRX2 inhibitors to eliminate acute and offending drug-induced pseudoallergic reactions.

## 7. Host Defense

MRGPRX2 plays an important role not only in relation to diseases but also as a biological defense mechanism in mast cells [138,139]. MCs are the most frequently found multifunctional immune cells at the interface between the host and the environment; thus, MCs function as defensive immune response cells that sense microbial attack [2]. It is known that MCs express a variety of receptors, including MRGPRX2, that allow them to recognize a variety of pathogenic stimuli [2]. Since hBDs and LL-37, small cationic antimicrobial peptides produced by epithelial cells, induce MC activation through MRGPRX2, MRGPRX2-mediated MC activation could contribute to the innate immune function of mast cells [28,29,39]. The activation of MrgprB2 by quorum-sensing peptides such as competence-stimulating peptide-1, which is a mediator of interbacterial communication, inhibits bacterial growth, prevents biofilm formation, and effectively eliminates bacteria by recruiting neutrophils [140]. Local mast cell activation via MRGPRX2 plays a role in eliminating bacterial infections of the skin, promoting healing and protecting against reinfection [138]. Therefore, MRGPRX2 inhibition may pose risks such as opportunistic infections. When developing therapeutic agents for the diseases described in the above section, it is necessary to develop compounds with less inhibitory activities against bacterial infection. We believe that the therapeutic effect of MRGPRX2 inhibition can be exploited to the maximum extent by lowering the risk of opportunistic infections.

## 8. MRGPRX2 Inhibitors

MRGPRX2 is thought to play important roles in the above-mentioned diseases by inducing neurogenic inflammation, and thus, MRGPRX2 inhibitors are expected to improve these diseases. MRGPRX2 inhibitors can be divided into three categories: (1) direct inhibition of MRGPRX2, (2) inhibition of MRGPRX2 downstream signaling, and (3) other mechanisms (Figure 2 and Table 1).

### 8.1. Direct Inhibition of MRGPRX2

QWF is a tripeptide antagonist of NK-1, a canonical receptor for SP which is a representative neuropeptide of the MRGPRX2 ligands. QWF has also been reported to act as an MRGPRX2 antagonist. QWF competitively inhibits SP binding not only to human MRGPRX2 but also to Mrgpra1 and b2 [36]. QWF inhibits SP-induced MC activation and ameliorates SP-induced pain that is not suppressed in NK-1-deficient mice [36]. From a therapeutic point of view, QWF lacks specificity and cannot eliminate the side effects caused by inhibitory effects other than MRGPRX2. Therefore, a specific antagonist for MRGPRX is required. Compounds 1 and 2, small molecule MRGPRX2 antagonists obtained by screening a small molecule compound library, inhibit MRGPRX2 activation by several MRGPRX2 ligands including SP and icatibant but show no inhibitory effect on NK-1 or other GPCRs. Compounds 1 and 2 inhibited the MC activation by MRGPRX2 ligand, and the intervention of both compounds revealed the involvement of MAPKs in the downstream signaling of MRGPRX2. However, compounds 1 and 2 did not show any inhibitory effect on SP-induced mouse MC activation [42]. The alkaloid compound piperine, reported as a small molecule MRGPRX2 antagonist, has a cross-inhibitory effect with mice and ameliorates mouse anaphylactic reactions [141]. In addition, naturally occurring compounds such as isoliquiritigenin (a component of licorice), shikonin (a component of Lithospermum erythrorhizon), imperatorin (an active furocumarin in *Angelica Dahurica radix*), and roxithromysin (a derivative of erythromycin) were shown to bind to MRGPRX2 in molecular docking studies and surface plasmon resonance (SPR) and inhibit C48/80 or SP-induced passive cutaneous anaphylaxis (PCA) in mice [50,142,143,144]. Paeoniflorin (a component of *Paeonia Lactiflora*), quercetin (a plant flavonoid), and genistein (a non-steroidal polyphenol), which were found to bind MRGPRX2 only in the docking model, also suppressed C48/80-induced PCA in mice [145,146,147]. On the other hand, naturally occurring components such as piperine, isoliquiritigenin, and shikonin are also known to act on various enzymes [141,160,161]. These compounds are low-affinity antagonists with inhibitory concentrations on the order of μM. These low-affinity MRGPRX2 antagonists remain a concern for side effects based on their non-MRGPRX2 effects. In order to evaluate the efficacy and pharmacokinetics of high-affinity MRGPRX2 antagonists, experimental animals are essential. Although mouse Mrgprb2 is considered to be the human MRGPRX2 ortholog, the homology between MRGPRX2 and Mrgprb2 is low and there are differences in the intensity of action of each ligand [20]. It has been suggested that MRGPRX2 has a different binding site for each ligand and more than one activation site to transduce the signal [162]. Since both human MRGPRX2 and mouse Mrgpb2 cover a wide range of ligands, it will be difficult to obtain common antagonists that inhibit all ligand binding with high affinity. However, it may be possible to acquire common high-affinity antagonists competing in binding with both receptors for a specific MRGPRX2 ligand that shows a pathology. Therefore, it might have a chance to develop a therapeutic agent with high affinity for MRGPRX2. The predictive model for the complex of ZINC-3573 and MRGPRX2 [163] has allowed several groups to perform docking studies [50,142,145,146,147]. These trials might contribute to create high-affinity MRGPRX2 antagonists if the predicted site is applicable for estimating the binding mode of interaction sites with MRGPRX2. Furthermore, it would be possible to blush up the first hit compounds, which have been generally obtained from library screening, and even the low activity compounds eliminated during the screening process. Several known MRGPRX2 antagonists should be assessed with docking studies as to whether or not they have a common binding mode in the predicted ligand site.

For the purpose of ameliorating acute drug-induced pseudoallergic reactions in MCs, it will be useful to develop high-affinity MRGPRX2 inhibitors for a wide range of MRGPRX2 ligands. DNA aptamer has been reported as a method to obtain high-affinity MRGPRX2 inhibitors [148]. Nucleic acid aptamer is a shape complement that binds to proteins by creating a three-dimensional structure that fits the shape of the protein. Suzuki et al. screened for single strand DNA (ssDNA) that binds MRGPRX2 using a proteoliposome that incorporates MRGPRX2 into a liposome to obtain a high-affinity MRGPRX2 antagonist, aptamer-X35 [148]. Because a nucleic acid aptamer fits the shape of proteins, it is difficult to obtain a nucleic acid aptamer that commonly recognizes between orthologs with low homology. Therefore, the attenuation of SP-induced anaphylactic reaction by aptamer-X35 has been confirmed using a model of MC-deficient rats transplanted with MRGPRX2-expressing MC lines [148]. It has also been reported that in vivo MRGPRX2 function can be assessed in a human hematopoietic stem cell engraftment model using mast cell-deficient mice (NOD-scid IL2R-γ-/- strain) [164]. Furthermore, a humanized mouse pathological model in which Mrgprb2 is deleted and replaced by human MRGPRX2 is also a promising way to effectively evaluate MRGPRX2 inhibitors for human disease improvement. However, considering that the mouse *Mrgpr* family forms clusters in the genome and the Mrgpr family shows common reactivity to ligands [20], it is necessary to determine whether the cluster region containing Mrgprb2 should be deleted for each pathology. It is also important to take into consideration the expression intensity and the localization of MRGPRX2 in humans.

### 8.2. Inhibition of MRGPRX2 Downstream Signaling

Recently, inhibitory compounds for MRGPR2 with anti-inflammatory properties by blocking the above-mentioned MRGPRX2 downstream signaling have been reported. Resveratrol, a polyphenol found in grapes, red wine, and peanuts, increases the expression of nuclear factor erythroid-derived 2-related factor (Nrf2) and induces heme-oxygenase 1 (HO-1), which has antioxidant effects [149]. Since the Nrf2/HO-1 pathway is known to suppress NF-κB signaling downstream of MRGPRX2 [150], it may indirectly suppress MRGPRX2 activation. Recently, it was reported that licochalcone A, a chalcone isolated from Glycyrrhiza uralensis Fisch, also suppresses MRGPRX2-mediated MC activation by inhibiting NF-κB signaling [151]. Licochalcone A has been reported to activate Nrf2 [152], so that licochalcone A may also suppress MRGPRX2 signaling via a similar mechanism to resveratrol. Osthole, a coumarin derivative extracted from the medicinal herbs of the Cnidium monnieri (L.) Cusson plant, does not interact with MRGPRX2 but inhibits MRGPRX2 ligand-stimulated increases in intracellular Ca^2+^ concentration and suppresses MC activation. It also suppresses MRGPRX2 expression on the plasma membrane, and thus, MRGPRX2 is inhibited through a complex mechanism [153]. In addition, Osthole has also been reported to bind to IgE receptors, which may indicate an anti-inflammatory effect of dual action on MCs [154]. In addition, dexamethasone, a well-known anti-inflammatory drug, inhibits MC activation through downregulating Gαi and MRGPRX2 downstream signals induced by basic secretagogues [155]. Lactic acid inhibits MRGPRX2 ligand-induced MC activation by suppressing MRGPRX2 downstream signals, such as elevation of intracellular Ca^2+^ concentration and activation of MAP kinase. However, it may also inhibit MRGPRX2 activation and MC activation by interacting with the basicity of MRGPRX2 ligands [156]. Therefore, the development of therapeutic agents targeting MRGPRX2 downstream signaling may also be effective on animal disease models.

### 8.3. Others

CD300f (leukocyte mono-immunoglobulin-like receptor 3), a known inhibitory regulator of IgE- and LPS-stimulated MC activation, suppresses MRGPRX2-mediated MC activation [165]. CD300f inhibits the activation of FcεR and TLR by binding to ceramide and sphingomyelin [157]. Although the mechanism by which CD300f regulates MRGPRX2 suppression is unknown, these results suggest that CD300f activators may be effective in suppressing anaphylactic reactions involving the MRGPRX2 pathway. Dondalska et al. reported that immunomodulatory single strand DNA (ssDNA) suppresses MRGPRX2-mediated MC activation using basic secretagogues. The ssDNA suppresses C48/80- and LL-37-induced murine dermatitis models [158]. In addition, sugammadex, a cyclodextrin derivative encapsulating the MRGPRX2 ligand rocuronium, inhibits MRGPRX2 activation and MC activation by not only rocuronium but also SP and other MRGPRX2 ligand stimuli [159]. These detailed mechanisms are awaiting further research.

## 9. Future Prospects

High-affinity MRGPRX2 inhibitors are expected to be applied to areas where there are few effective therapeutic agents, such as postoperative pain, migraine, and drug-induced acute pseudoallergic reactions. The key is to obtain a specific antagonist with high affinity for MRGPRX2, which has the peculiar ligand recognition ability. Although molecular modeling experiments have enabled us to predict the 3D structure of MRGPRX2 for designing agonists [163], the exact 3D structure of MRGPRX2 in complexes with ligands and other inhibitory compounds has not yet been elucidated. The crystal structures of more than 50 GPCRs have been clarified in the past decade, and it has become possible to solve the structure of GPCR complexes through technological innovations in cryo-electron microscopy (cryoEM) and NMR [166]. In the future, the combination of 3D structure analysis of MRGPRX2-agonist/antagonist complexes using X-ray crystallography, cryoEM, and NMR will enable structure-based rational design of high-affinity MRGPRX2 binding compounds. The development of a specific antagonist with high affinity for MRGPRX2 will provide a deeper understanding not only of the activation mechanism of MRGPRX2, but also of various diseases of neurogenic inflammation and pseudoallergic reactions. The elucidating mechanism of the unparalleled GPCR could lead to medicinal developments for ameliorating unresolved diseases and side effects from drugs associated with this receptor.

So far, rodents have made significant contributions to the elucidation of the MRGPX2 function in vivo. However, the results from the Mrgprb2-KO mouse may be re-examined, because, as mentioned above, the Mrgpr family of receptors may compensate for the lost function. The rat-sized monkey or common marmoset (Callithrix jacchus) is becoming an attractive animal model in biomedical research [167]. It may be available as a non-human primate with 85% MRGPRX2 homology. In addition, it is possible to perform behavioral analyses that may not be observed in rodents. Recently, it has become possible to easily genetically modify marmosets, so that they can be used for neuroscience research [168]. In the future, it is expected that animal models closer to that of humans than rodents will be used to analyze MRGPRX2-related pathology.

## 10. Conclusions

The development of high-affinity MRGPRX2 inhibitors is expected to make a significant contribution to the treatment of neurogenic inflammation, type 2 inflammation such as atopic dermatitis and chronic urticaria, and non-histaminic itch. They may also be a useful drug for unexpected, drug-induced pseudoallergic reactions. It will be necessary, however, to find a way to preserve the mast cell defense mechanism of MRGPRX2.

## Figures and Tables

**Figure 1 cells-10-02906-f001:**
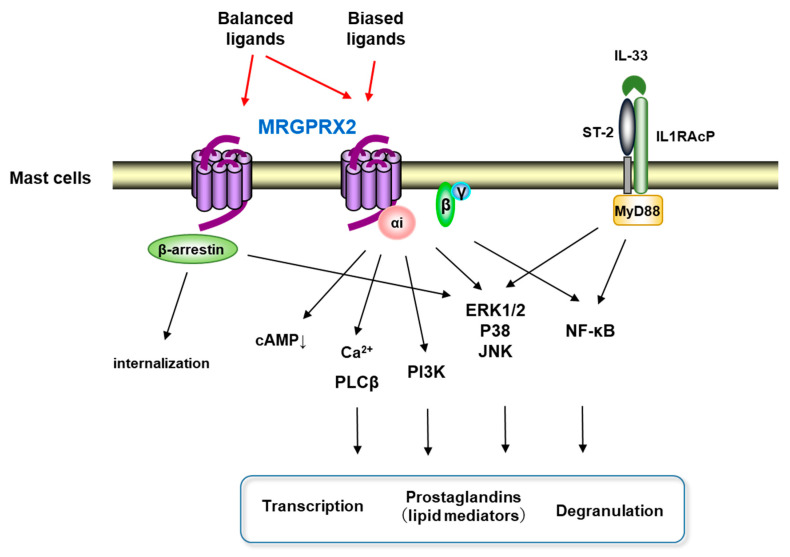
Scheme of the signal pathway after MGPRX2 activation. MRGPRX2 ligands activate Gi protein, which induces a cAMP decrease, Ca^2+^ mobilization, and also activates MAPKs, PI3K, and NF-κB pathways, which lead to degranulation, induction of transcription, and production of prostaglandins. Balanced MRGPRX2 ligands not only activate Gi protein but also recruit β-arrestin to induce internalization, while biased MRGPRX2 ligands cannot recruit β-arrestin. Additionally, IL-33 enhances MRGPRX2 signaling by activating MAPKs and NF-κB.

**Figure 2 cells-10-02906-f002:**
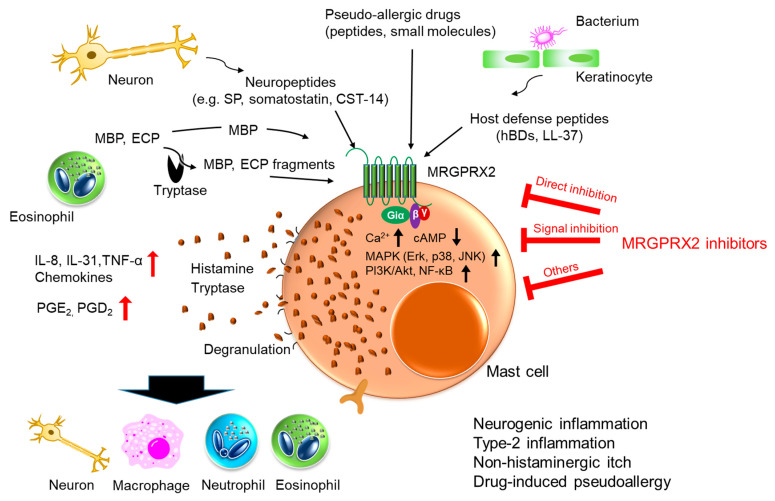
Schematic diagram of MRGPRX2 activation mediated by various stimuli and potential effects of MRGPRX2 inhibitors for induced diseases. MRGPRX2 is stimulated with various ligands and is thought to play important roles in neurogenic inflammation, type 2 inflammation, non-histaminergic itch, and drug-induced pseudoallergy through mast cell activation. MRGPRX2 inhibitors, which are divided into three categories, 1) direct inhibition of MRGPRX2, 2) inhibition of MRGPRX2 downstream signaling, and 3) other mechanisms, are expected to improve these diseases.

**Table 1 cells-10-02906-t001:** Inhibitors.

Type	Inhibitor	Mechanism	References
Direct inhibition	QWF	Dual action to MRGPRX2 and NK-1 receptor Competitive inhibition of SP binding to MRGPRX2 Inhibition of intracellular Ca^2+^ mobilization and mast cell degranulation Reduction of SP-induced itch	[36]
	Compound 1, 2	MRGPRX2 antagonist, not for NK-1 or M2R Competitive inhibition of SP binding to MRGPRX2 Inhibition of intracellular Ca^2+^ mobilization, ERK signaling, GTP binding, MC degranulation, and de novo PGD_2_ production	[32,42]
	Piperine	Direct interaction to MRGPRX2 determined by binding to MRGPRX2-expressing cell membrane Inhibition of intracellular Ca^2+^ mobilization, degranulation, histamine release, cytokines release, PLCγ1, PKC, inositol 1,4,5-triphate receptor, p38, PKB, and ERK Reduction of C48/80-induced anaphylactoid reactions	[141]
	Isoliquiritigenin	Predicted direct interaction to MRGPRX2 Binding to MRGPRX2 by molecular docking assay Inhibition of intracellular Ca^2+^ mobilization, cytokines release, MC degranulation, and histamine release Reduction of C48/80-induced anaphylactoid reactions	[142]
	Shikonin	Direct interaction to MRGPRX2 determined by surface plasmon resonance and molecular docking analysis Inhibition of intracellular Ca^2+^ mobilization, MC degranulation, histamine release, cytokines release, PLCγ1, PKC, inositol 1,4,5-triphate receptor, and ERK Reduction of C48/80-induced anaphylactoid reaction	[143]
	Imperatorin	Direct interaction to MRGPRX2 determined by surface plasmon resonance and molecular docking analysis Inhibition of intracellular Ca^2+^ mobilization, MC degranulation, histamine release, cytokines release, CamKII, and ERK Reduction of SP-induced anaphylactoid reaction and OVA-induced lung inflammtion	[50]
	Roxithromysin	Direct interaction to MRGPRX2 by surface plasmon resonance Inhibition of intracellular Ca^2+^ mobilization, MC degranulation, histamine release, cytokines release, PLCγ1, inositol 1,4,5-triphate receptor, and p38 Reduction of C48/80-induced anaphylactoid reaction	[144]
	Paeoniflorin	Predicted direct interaction to MRGPRX2 Binding to MRGPRX2 by molecular docking analysis Inhibition of intracellular Ca^2+^ mobilization, MC degranulation, histamine release, cytokines release, PLCγ1, inositol 1,4,5-triphate receptor, p38, ERK, AKT, and PKC Reduction of C48/80-induced anaphylactoid reaction	[145]
	Quercetin	Predicted direct interaction to MRGPRX2 Binding to MRGPRX2 by molecular docking analysis Inhibition of intracellular Ca^2+^ mobilization, MC degranulation, histamine release, cytokines release, PLCγ1, inositol 1,4,5-triphate receptor, and ERK Reduction of C48/80-induced anaphylactoid reaction	[146]
	Genistein	Predicted direct interaction to MRGPRX2 Binding to MRGPRX2 by molecular docking analysis Inhibition of intracellular Ca^2+^ mobilization, β-arrestin recruitment, and MC degranulation Reduction of C48/80-induced anaphylactoid reaction	[147]
	Aptamer-X35	ssDNA aptamer Direct interaction to MRGPRX2 determined by binding to MRGPRX2 expressing cells Inhibition of MC degranulation and histamine release Reduction of SP-induced anaphylactoid reaction in MRGPRX2-expressing cell engraft rat	[148]
MRGPRX2 signal inhibitor	Resveratrol	Inhibition of MRGPRX2 downstream signal, NF-κB via Nrf2/HO-1 pathway activation Inhibition of intracellular Ca^2+^ mobilization, MC degranulation, histamine release, and cytokines release Reduction of C48/80-induced anaphylactoid reaction	[149,150]
	Licochalcone A	Inhibition of MRGPRX2 downstream signal, NF-κB Inhibition of intracellular Ca^2+^ mobilization, MC degranulation, and cytokines release Reduction of SP-induced anaphylactoid reaction	[151,152]
	Osthole	Inhibition of MRGPRX2-mediated intracellular Ca^2+^ mobilization, ERK, MC degranulation, and cytokines release Reduction of C48/80-induced paw edema and LL-37-induced inflammation in vivo	[153]
	Dexamethasone	Downregulation of Gαi and MRGPRX2 downstream signals	[154]
	Lactic acid	Inhibition of MRGPRX2-mediated intracellular Ca^2+^ mobilization, MC degranulation, and cytokines release Reduction of LL-37-induced inflammation in vivo	[155]
Others	Ceramide, sphingomyelin	Inhibition of MRGPRX2-mediated mast cell activation by CD300f activation Reduction of MRGPRX2 ligands-induced inflammation in vivo	[156,157]
	ssDNA	Unknown mechanism Reduction of MRGPRX2 ligands-induced inflammation in vivo	[158]
	Sugammadex	Encapsulation of MRGPRX2 ligand Inhibition of intracellular Ca^2+^ mobilization, MC degranulation, and CCL2 release	[159]

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
