# Peer review of "Therapeutic Potential of MRGPRX2 Inhibitors on Mast Cells"

_cells, 2021, doi:10.3390/cells10112906_

Round 1
Reviewer 1 Report
In this paper, the authors review some entities that develop inflammation, in which the mast-cell receptor MRGPRX2 could have a role and therefore the inhibitors of such receptor are described as a potential treatment for these diseases.
- Even though this is a review on the role of this receptor in the inflammatory diseases, the authors have included some entities in which the role of the receptor has not been proven yet (for example in migraine), or either the usefulness of these treatments could be considered as doubtful (for example in contact dermatitis). And on the contrary they have not mentioned other highly prevalent inflammatory diseases in which a new treatment would be useful (chronic urticaria or bronchial asthma). The authors should explain which criteria they used to select the pathologies in which the inhibition of this receptor could play an important role.
- It is interesting the section where the authors briefly describe the substances that are inhibitors of the receptor. Some substances have been included in the table but are not described in the text.
Minor comments:
-Line 48-51: This sentence is too long and difficult to understand.
- I did not find in the text the meaning of CGRP (calcitonine gene related peptide).
- Line 263: excessive repetition that AD is an inflammatory disease.
- L303-4: the fact that the inflammation is decreased in a model of contact dermatitis and this includes anaphylactic reactions is a contradiction. Contact dermatitis is a model of type IV delayed hypersensitivity and anaphylactic reactions are the prototype of immediate reactions (that can be IgE-mediated or not).
- L375: codeine is used as antitussive or as analgesic, not to treat schizophrenia.
- L390-403: long and complicated explanation on the canine receptor MRGPRX2, that could be eliminated considering that the authors admit that its results cannot be extrapolated to the human receptor.
Author Response
Replies to the Reviewer 1
Comments by the Reviewer #1:
In this paper, the authors review some entities that develop inflammation, in which the mast-cell receptor MRGPRX2 could have a role and therefore the inhibitors of such receptor are described as a potential treatment for these diseases.
Reply to the comments by the reviewer #1
We deeply appreciate the reviewer’s careful reviewing and positive comments.
Comments by the Reviewer #1:
- Even though this is a review on the role of this receptor in the inflammatory diseases, the authors have included some entities in which the role of the receptor has not been proven yet (for example in migraine), or either the usefulness of these treatments could be considered as doubtful (for example in contact dermatitis). And on the contrary they have not mentioned other highly prevalent inflammatory diseases in which a new treatment would be useful (chronic urticaria or bronchial asthma). The authors should explain which criteria they used to select the pathologies in which the inhibition of this receptor could play an important role.
Reply to the comments by the reviewer #1
Thank you for your suggestion. We composed this article to discuss the broad potential of MRGPRX2 inhibitors as therapeutic agents. We hope you will understand our intent. I would also like to add a missing sub-section and discuss chronic urticaria, for which we would anticipate therapeutic effects as follows:
“4.2 Chronic urticaria (CU)
In CU, in which wheals occur daily over the entire body for at least 6 weeks, the release of various vasoactive autacoid mediators, proteases, chemokines, and cytokines from MCs plays an important role in inducing the wheal response. Approximately 40% of patients with chronic spontaneous urticaria have circulating IgE or FcεR1-recognizing IgG antibodies, and 30-50% of patients with CU express high levels of FcεRI. Around half the patients with CU the condition has been implicated in IgE and autoimmune mechanisms, and the remainder are considered to have idiopathic CU. On the other hand, it has been reported that there is no change in the number of mast cells in the lesions but there is a higher percentage of MRGPRX2-positive mast cells in CU patients compared to healthy subjects. In addition, intradermal injection of neuropeptides such as MRGPRX2 ligand SP and vasoactive intestinal peptide (VIP) enhances the wheal response in CU patients in coparison with healthy subjects. Compared to healthy individuals, MRGPRX2 expressing mast cells and MRGPRX2 ligands such as SP in serum derived from CU patients are upregulated. Similar to AD lesions, infiltration with eosinophils and prominent deposition of MBP and eosinophil cationic protein (ECP) have been reported in CU lesions. MBP and MBP and ECP tryptase fragments activate MRGPRX2 to activate mast cells. These reports suggest the involvement of MRGPRX2 in mast cells in CU lesions.”
Comments by the Reviewer #1:
- It is interesting the section where the authors briefly describe the substances that are inhibitors of the receptor. Some substances have been included in the table but are not described in the text.
Reply to the comments by the reviewer #1
Thank you for pointing this out. we have revised it as follows.
“In addition, naturally occurring compounds such as isoliquiritigenin (a component of licorice, shikonin (a component of Lithospermum erythrorhizon), Imperatorin (an active furocumarin in Angelica Dahurica radix) and Roxithromysin (a derivative of erythromycin) were shown to bind to MRGPRX2 in molecular docking studies and surface plasmon resonance (SPR) and inhibit C48/80 or SP-induced passive cutaneous anaphylaxis (PCA) in mice. Paeoniflorin (a component of Paeonia Lactiflora) and quercetin (a plant flavonoid), which were found to bind MRGPRX2 only in the docking model, also suppressed C48/80-induced PCA in mice.” (Line 474-479)
Comments by the Reviewer #1:
Minor comments:
-Line 48-51: This sentence is too long and difficult to understand.
Reply to the comments by the reviewer #1
Thank you for pointing this out. we have revised it as follows.
“It is known that MCs are activated by basic secretagogues such as substance P (SP) and compound 48/80 (C48/80), as well as through suppression of tumorigenicity-2 as an IL-1 family receptor for IL-33 and a receptor for thymic stromal lymphopoietin.” (Line 51-54)
Comments by the Reviewer #1:
- I did not find in the text the meaning of CGRP (calcitonine gene related peptide).
Reply to the comments by the reviewer #1
We apologize for the mistake and have removed the description of CGRP.
Comments by the Reviewer #1:
- Line 263: excessive repetition that AD is an inflammatory disease.
Reply to the comments by the reviewer #1
Thank you for your suggestion, we have removed the sentence “AD can be divided into two subtypes: extrinsic AD and intrinsic AD. Extrinsic AD is an IgE-dependent disease correlated with high serum IgE levels, as with rhinitis and asthma. In contrast, intrinsic AD is characterized by the absence of allergen-specific IgE.”
Comments by the Reviewer #1:
- L303-4: the fact that the inflammation is decreased in a model of contact dermatitis and this includes anaphylactic reactions is a contradiction. Contact dermatitis is a model of type IV delayed hypersensitivity and anaphylactic reactions are the prototype of immediate reactions (that can be IgE-mediated or not).
Reply to the comments by the reviewer #1
As you pointed out, I apologize for the wrong information. we have revised it as follows.
“Mrgprb2-KO mice have been reported to have significantly reduced inflammation, including itching in various models of allergic contact dermatitis, suggesting that MRGPRX2 is involved in the development of ACD as well as atopic dermatitis.” (Line 325-328)
Comments by the Reviewer #1:
- L375: codeine is used as antitussive or as analgesic, not to treat schizophrenia.
Reply to the comments by the reviewer #1
We apologize for stating the wrong information. we have revised it as follows.
“Subsequently, clozapine, an antipsychotic drug, and codeine have been reported to cause pseudoallergic reactions via MRGPRX2.” (Line 398-400)
Comments by the Reviewer #1:
- L390-403: long and complicated explanation on the canine receptor MRGPRX2, that could be eliminated considering that the authors admit that its results cannot be extrapolated to the human receptor.
Reply to the comments by the reviewer #1
As suggested, we removed description of reactivity of canine MRGPRX2 in Line390-403 “Recently, canine MRGPRX2 has been identified and reported to be more responsive to MRGPRX2 ligands than human MRGPRX2. Considering that MRGPRX2 ligand reactivity is relatively blunted in mouse Mrgprb2 compared to human MRGPRX2, this is consistent with the species differences in drug-induced pseudoallergic reactivity. It is difficult to extrapolate the pseudo-allergic properties of drugs in humans in evaluations using dogs, because not all drugs causing MRGPRX2-mediated MC pseudoallergic reactions in dogs also induce the reactions in humans.”
Reviewer 2 Report
The article represents a comprehensive narrative review about therapeutic potential of MRGPRX2 inhibitors on mast cells, indexing more than 150 references.
I would recommend that the authors describe the material and methods used (criteria of literature search - terms, period of time and databases used).
Also, I would recommend that the article should finish with some concise conclusions.
Author Response
Replies to the Reviewer #2
Comments by the Reviewer #2:
The article represents a comprehensive narrative review about therapeutic potential of MRGPRX2 inhibitors on mast cells, indexing more than 150 references.
Reply to the comments by the reviewer #2
We deeply appreciate the reviewer’s careful review and positive comments.
Comments by the Reviewer #2:
I would recommend that the authors describe the material and methods used (criteria of literature search - terms, period of time and databases used).
Reply to the comments by the reviewer #2
We deeply appreciate the reviewer’s careful reviewing and positive comments.
For about 15 years, we have been focusing on MRGPRX2 for drug development. During this time, we have been keeping a close eye on the latest information on empirically applicable diseases and MRGPRX2 inhibitors. We have prepared this paper based on the accumulated literature that we have collected without utilizing mechanical bioinformatics or big databases. Therefore, there are no contents to be described as the material and methods.
Comments by the Reviewer #2:
Also, I would recommend that the article should finish with some concise conclusions.
Reply to the comments by the reviewer #2
We deeply appreciate the constructive comments by the reviewer #2.
As suggested, we added a section as follows:
“10. Conclusion
The development of high affinity MRGPRX2 inhibitors is expected to make a significant contribution to the treatment of neurogenic inflammation, type 2 inflammation such as atopic dermatitis and chronic urticaria, and non-histaminic itch. It may also be as a useful drug for unexpected drug-induced pseudoallergic reactions. It will be necessary, however, to find a way to preserve the mast cell defense mechanism of MRGPRX2.”
Reviewer 3 Report
In this review, authors present available information related to the function of MRGPRX2 in mast cells, discussing the role of this receptor on different MC-related inflammatory reactions. It is a comprehensive and detailed review that certainly present information in a systematic form and importantly contributes to the comprehension of the importance of this recently identified receptor in MCs with unexpected relevant functions.
Some aspects could be improved.
Major comments:
- In the abstract and in the initial paragraphs it is mentioned that there are two major pathways of MC activation (IgE-dependent and IgE-independent). Authors mention that the IgE-dependent pathway has been widely studied but the independent one remains to be fully explored. Recent transcriptomic studies showing the expression of multiple receptors on MCs coupled to distinct secretory pathways (anaphylactic, piecemeal and constitutive), indicate that there is not only one, but many IgE-independent activation pathways in MCs. This should be clarified in the text.
- Despite the detailed description of the ligands and binding characteristics of substances interacting MRGPRX2, it is stated that “the extracellular region of the GPCR binds the ligand, while the intracellular region is involved in the binding of G proteins” (lines 140-141). The reference mentioned there (39) is a review on the importance and therapeutic potential of biased ligands of GPCRs. In that review, no specific mention on the binding site of GPCR ligands is made, since it is known that, in general, binding occurs in a hydrophobic pocket formed by the transmembrane segments of the receptors. Furthermore, it has been shown that naturally occurring missense MRGPRX2 variants affecting mast cell degranulation are located in aminoacids located at transmembrane helixes of the receptor (Alkanfari, I., Gupta, K., Jahan, T. and Ali, H. J Immunol, 201:343-349, 2018). This must be mentioned in the corresponding section.
- Figure 1 shows a scheme on the signaling pathways associated to MRGPRX2 receptor. However, as in other GPCRs, it is expected that beta-arrestin not only regulate receptor internalization, but also intracellular signaling (as mentioned in reference 39). Please indicate this in the scheme.
- Figure 2 shows the multiple ligands for MRGPRX2, but also shows in red lines the actions of described inhibitors. Please include the inhibitors on Figure 2 title.
- Table I mentions that Piperine, Isoliquiritigenin, Shikokin, Imperatonin, Roxithromysin, Paeoniflorin and Quercetin have been proposed to bind MRGPRX2 by molecular docking analysis or surface plasmon resonance, but references included also mention an inhibitory effect of those compounds on distinct MC-related reactions and, in some cases, a possible mechanism of inhibition is mentioned. That important information must be included in Table I, as has been done with other compounds.
- This interesting review finishes making emphasis on the importance of searching inhibitors of MRGPRX2 for therapeutic use. However, due to the relevance of this receptor in the control of inflammatory mediator release, and its underscored role on protective reactions, possible non-desirable effects of those compounds should be mentioned.
- In line with the previous comment, few lines speculating on the possible physiological role of MRGPRX2 in mas cells could improve the review.
Minor comments:
- Line 52 states “MC activations by basic”. Please correct to “activation”
- Line 62 states “expresses”. Please substitute by “expressed”
- Please correct calcinurin by calcineurin on line 147.
- Please clarify what is meant with the expression “specially superior…” on line 187.
- Figure 2 shows FcεR, please substitute by FcεRI
- Table I has no title. Please add one.
- Please clarify what is meant with the sentence “The method of replacing mouse MCs with human MCs as rats has also been reported” (line 474)
Author Response
Replies to the Reviewer #3
Comments by the Reviewer #3:
In this review, authors present available information related to the function of MRGPRX2 in mast cells, discussing the role of this receptor on different MC-related inflammatory reactions. It is a comprehensive and detailed review that certainly present information in a systematic form and importantly contributes to the comprehension of the importance of this recently identified receptor in MCs with unexpected relevant functions.
Some aspects could be improved.
Reply to the comments by the reviewer #3
We deeply appreciate the reviewer’s careful reviewing and positive comments.
Comments by the Reviewer #3:
Major comments:
In the abstract and in the initial paragraphs it is mentioned that there are two major pathways of MC activation (IgE-dependent and IgE-independent). Authors mention that the IgE-dependent pathway has been widely studied but the independent one remains to be fully explored. Recent transcriptomic studies showing the expression of multiple receptors on MCs coupled to distinct secretory pathways (anaphylactic, piecemeal and constitutive), indicate that there is not only one, but many IgE-independent activation pathways in MCs. This should be clarified in the text.
Reply to the comments by the reviewer #3
Thank you very much for the important comments and suggestions.
As suggested, the sentence in the abstract " Although the mechanism of IgE-dependent MC activation has been well studied, the IgE-independent pathway remains unclear." has been revised as follows; “Although IgE-dependent signaling is the main pathway to MC activation, the IgE-independent pathways have also been found to serve pivotal roles in the pathophysiology of various inflammatory conditions. Recent studies have shown that human and mouse MCs express several regulatory receptors such as toll-like receptors (TLRs), CD48, C300a and GPCRs including mas-related GPCR-X2 (MRGPRX2).”
As suggested, the sentence on line 46-47 “Although the mechanism of IgE-dependent MC activation has been well studied, the IgE-independent pathway remains unresolved.” has been revised as follows; “Although the IgE/FcεRI signaling is the main pathway for MC activation, the ability of mast cells to alter their responses to so many internal response to so many internal and external signals suggests that the various regulatory receptors such as toll-like receptors (TLRs), CD48, CD300a, lectin receptors and GPCRs also act as regulators of these responses.” (Line 47-51)
Comments by the Reviewer #3:
Despite the detailed description of the ligands and binding characteristics of substances interacting MRGPRX2, it is stated that “the extracellular region of the GPCR binds the ligand, while the intracellular region is involved in the binding of G proteins” (lines 140-141). The reference mentioned there (39) is a review on the importance and therapeutic potential of biased ligands of GPCRs. In that review, no specific mention on the binding site of GPCR ligands is made, since it is known that, in general, binding occurs in a hydrophobic pocket formed by the transmembrane segments of the receptors. Furthermore, it has been shown that naturally occurring missense MRGPRX2 variants affecting mast cell degranulation are located in amino acids located at transmembrane helixes of the receptor (Alkanfari, I., Gupta, K., Jahan, T. and Ali, H. J Immunol, 201:343-349, 2018). This must be mentioned in the corresponding section.
Reply to the comments by the reviewer #3
Thank you very much for the important comments and suggestions.
As suggested, the sentence on line 140-141 “The extracellular region of the GPCR binds the ligand, while the intracellular region is involved in the binding of heterotrimeric G proteins, β-arrestin, and other downstream effectors” has been revised as follows; “It is generally known that GPCR ligands act on the hydrophobic pocket formed by the extracellular loops and the adjacent transmembrane regions of the GPCRs, converting them into cellular responses via G-proteins, β-arrestins and other downstream effectors. Furthermore, it has been shown that naturally occurring missense MRGPRX2 variants (G165E, D184H, W243R and H259Y) affecting mast cell degranulation are located in amino acids located at transmembrane helixes of the receptor.” (Line 143-148)
Comments by the Reviewer #3:
Figure 1 shows a scheme on the signaling pathways associated to MRGPRX2 receptor. However, as in other GPCRs, it is expected that beta-arrestin not only regulate receptor internalization, but also intracellular signaling (as mentioned in reference 39). Please indicate this in the scheme.
Reply to the comments by the reviewer #3
Thank you very much for the important comments and suggestions.
We indicated intracellular signaling by beta-arrestin in Figure 1 as suggested. The intracellular signaling has also been added to the text as follows.
“In addition to G protein signaling, C48/80 or SP recruits β-arrestin 1 via MRGPRX2 to internalize and desensitize MRGPRX2 at the plasma membrane, and to activate the ERK signaling pathway.” (Line 175-176)
Comments by the Reviewer #3:
Figure 2 shows the multiple ligands for MRGPRX2, but also shows in red lines the actions of described inhibitors. Please include the inhibitors on Figure 2 title.
Reply to the comments by the reviewer #3
Thank you very much for the important comments and suggestions.
As suggested, we revised the title of Figure 2 as follow; “Schematic diagram of MRGPRX2 activation mediated by various stimuli, and potential effects of MRGPRX2 inhibitors for induced diseases.”
Comments by the Reviewer #3:
Table I mentions that Piperine, Isoliquiritigenin, Shikokin, Imperatonin, Roxithromysin, Paeoniflorin and Quercetin have been proposed to bind MRGPRX2 by molecular docking analysis or surface plasmon resonance, but references included also mention an inhibitory effect of those compounds on distinct MC-related reactions and, in some cases, a possible mechanism of inhibition is mentioned. That important information must be included in Table I, as has been done with other compounds.
Reply to the comments by the reviewer #3
Thank you very much for the important comments and suggestions.
The results shown in the papers reporting each compound are stated in the Table.
Comments by the Reviewer #3:
This interesting review finishes making emphasis on the importance of searching inhibitors of MRGPRX2 for therapeutic use. However, due to the relevance of this receptor in the control of inflammatory mediator release, and its underscored role on protective reactions, possible non-desirable effects of those compounds should be mentioned.
In line with the previous comment, few lines speculating on the possible physiological role of MRGPRX2 in mast cells could improve the review.
Reply to the comments by the reviewer #3
Thank you very much for the important comments and suggestions.
As suggested, we added a section as follows:
“7. Host defense
MRGPRX2 plays an important role not only in relation to diseases but also as a biological defense mechanism in mast cells. MCs are the most frequently found multifunctional immune cells at the interface between the host and the environment, thus MCs function as defensive immune response cells that sense microbial attack. It is known that MCs express a variety of receptors, including MRGPRX2, that allow them to recognize a variety of pathogenic stimuli. Since hBDs and LL-37, small cationic antimicrobial peptides produced by epithelial cells induce MC activation through MRGPRX2, MRGPRX2-mediated MC activation could contribute to the innate immune function of mast cells. The activation of MrgprB2 by quorum-sensing peptides such as competence-stimulating peptide-1, which is a mediator of interbacterial communication, inhibits bacterial growth, prevents biofilm formation, and effectively eliminates bacteria by recruiting neutrophils. Local mast cell activation via MRGPRX2 plays a role in eliminating bacterial infections of the skin, promoting healing, and protecting against reinfection. Therefore, MRGPRX2 inhibition may pose risks such as opportunistic infections. When developing therapeutic agents for the diseases described in the above section, it is necessary to develop compounds with less inhibitory activities against bacterial infection. We believe that the therapeutic effect of MRGPRX2 inhibition can be exploited to the maximum extent by lowering the risk of opportunistic infections.”
Minor comments:
Comments by the Reviewer #3:
Line 52 states “MC activations by basic”. Please correct to “activation”
Reply to the comments by the reviewer #3
Thank you very much for careful reading. We corrected the terms as suggested.
Comments by the Reviewer #3:
Line 62 states “expresses”. Please substitute by “expressed”
Reply to the comments by the reviewer #3
Thank you very much for careful reading. We corrected the terms as suggested.
Comments by the Reviewer #3:
Please correct calcinurin by calcineurin on line 147.
Reply to the comments by the reviewer #3
Thank you very much for careful reading. We corrected the terms as suggested.
Comments by the Reviewer #3:
Please clarify what is meant with the expression “specially superior…” on line 187.
Reply to the comments by the reviewer #3
Thank you very much for careful reading. The term “specially superior…” on line 187 has been revise as follows: “MCs are predominantly located in close proximity to peripheral nerve endings compared to other innate immune cells, making them the first cells to respond to sensory nerve activation. MCs are also involved in the mobilization of a variety of innate immune cells, furthering the inflammatory cascade and sensitization of peripheral afferents.” (Line 195-198)
Comments by the Reviewer #3:
Figure 2 shows FcεR, please substitute by FcεRI
Reply to the comments by the reviewer #3
Thank you very much for careful reading. We corrected the terms as suggested.
Comments by the Reviewer #3:
Table I has no title. Please add one.
Reply to the comments by the reviewer #3
Thank you very much for careful reading. We added the title of Table.
Comments by the Reviewer #3:
Please clarify what is meant with the sentence “The method of replacing mouse MCs with human MCs as rats has also been reported” (line 474)
Reply to the comments by the reviewer #3.
We apologize for the confusing wording. It has been corrected as follows; “It has also been reported that in vivo MRGPRX2 function can be assessed in a human hematopoietic stem cell engraftment model using mast cell-deficient mice (NOD-scid IL2R-γ-/- strain).” (Line 516-518)
Reviewer 4 Report
Although there is partial overlap with recent reviews (e.g., Thapallya et al. in Current Allergy and Asthma Reports (2021) 21:3, published online on 4 January 2021), the review by Ogasawara brings some additional aspects which justify publication. However, other reviews re. MRGPRX2 published in 2021 should be mentioned and cited in the manuscript.
In Abstract (line 10-12) and other parts of the review, the authors write that two main pathways for mast cell activation are known: IgE-dependent and IgE-independent. At present, this statement is not correct because Ig-independent pathways are numerous ones and some of them have been extensively studied (see, e.g., Harvima et al., JACI 2014; 134: 530-44). Thus, the sentence“…the IgE-independent pathway remains unresolved“ (line 46-47) and several others (e.g., line 344-345, 352) should be modified.
Author Response
Replies to the Reviewer #4
Comments by the Reviewer #4:
Although there is partial overlap with recent reviews (e.g., Thapallya et al. in Current Allergy and Asthma Reports (2021) 21:3, published online on 4 January 2021), the review by Ogasawara brings some additional aspects which justify publication. However, other reviews are MRGPRX2 published in 2021 should be mentioned and cited in the manuscript.
Reply to the comments by the reviewer #4
We deeply appreciate the reviewer’s careful reviewing and positive comments.
We have cited the following paper reported in 2021.
Thapaliya, M.; Chompunud Na Ayudhya, C.; Amponnawarat, A.; Roy, S.; Ali, H. Mast Cell-Specific MRGPRX2: a Key Modulator of Neuro-Immune Interaction in Allergic Diseases. Curr. Allergy Asthma Rep. 2021, 21, 3. doi: 10.1007/s11882-020-00979-5.
Kumar, M.; Duraisamy, K.; Chow, B.K. Unlocking the Non-IgE-Mediated Pseudo-Allergic Reaction Puzzle with Mas-Related G-Protein Coupled Receptor Member X2 (MRGPRX2). Cells 2021, 10, 1033. doi: 10.3390/cells10051033.
Ali, H. Revisiting the role of MRGPRX2 on hypersensitivity reactions to neuromuscular blocking drugs. Curr. Opin. Immunol. 2021, 72, 65-71. doi: 10.1016/j.coi.2021.03.011.
Roy, S.; Chompunud Na Ayudhya, C.; Thapaliya, M.; Deepak, V.; Ali, H. Multifaceted MRGPRX2: New insight into the role of mast cells in health and disease. J. Allergy Clin. Immunol. 2021, 148, 293-308. doi: 10.1016/j.jaci.2021.03.049.
Comments by the Reviewer #4:
In Abstract (line 10-12) and other parts of the review, the authors write that two main pathways for mast cell activation are known: IgE-dependent and IgE-independent. At present, this statement is not correct because Ig-independent pathways are numerous ones and some of them have been extensively studied (see, e.g., Harvima et al., JACI 2014; 134: 530-44). Thus, the sentence“…the IgE-independent pathway remains unresolved“ (line 46-47) and several others (e.g., line 344-345, 352) should be modified.
Reply to the comments by the reviewer #4
Thank you very much for the important comments and suggestions.
As suggested, the sentence in the abstract " Although the mechanism of IgE-dependent MC activation has been well studied, the IgE-independent pathway remains unclear." has been revised as follows; “Although IgE-dependent signaling is the main pathway to MC activation, the IgE-independent pathways also serve pivotal roles in the pathophysiology of various inflammatory conditions. Recent studies have shown that human and mice MCs express several regulatory receptors such as toll-like receptors (TLRs), CD48, C300a and GPCRs including mas-related GPCR-X2 (MRGPRX2).”
As suggested, the sentence in text line 46-47 “Although the mechanism of IgE-dependent MC activation has been well studied [8, 9], the IgE-independent pathway remains unresolved.” has been revised as follow; “Although IgE/FcεRI signaling is the main pathway for MC activation, the ability of mast cells to alter their responses to so many internal response to so many internal and external signals suggests that the various regulatory receptors such as toll-like receptors (TLRs), CD48, CD300a, lectin receptors and GPCRs also acts as a regulator of these response.”
In addition, "non-IgE" was changed to "SP" to accurately reflect the references cited in lines 368-376.
Round 2
Reviewer 1 Report
Thank you for your revision